# Study on the Strip Warpage Issues Encountered in the Flip-Chip Process

**DOI:** 10.3390/ma15010323

**Published:** 2022-01-03

**Authors:** Wan-Chun Chuang, Wei-Long Chen

**Affiliations:** Department of Mechanical and Electromechanical Engineering, Engineering Technology Research & Promotion Center, National Sun Yat-sen University, Kaohsiung 804, Taiwan; d043020007@student.nsysu.edu.tw

**Keywords:** flip-chip process, strip warpage, bump

## Abstract

This study successfully established a strip warpage simulation model of the flip-chip process and investigated the effects of structural design and process (molding, post-mold curing, pretreatment, and ball mounting) on strip warpage. The errors between simulated and experimental values were found to be less than 8%. Taguchi analysis was employed to identify the key factors affecting strip warpage, which were discovered to be die thickness and substrate thickness, followed by mold compound thickness and molding temperature. Although a greater die thickness and mold compound thickness reduce the strip warpage, they also substantially increase the overall strip thickness. To overcome this problem, design criteria are proposed, with the neutral axis of the strip structure located on the bump. The results obtained using the criteria revealed that the strip warpage and overall strip thickness are effectively reduced. In summary, the proposed model can be used to evaluate the effect of structural design and process parameters on strip warpage and can provide strip design guidelines for reducing the amount of strip warpage and meeting the requirements for light, thin, and short chips on the production line. In addition, the proposed guidelines can accelerate the product development cycle and improve product quality with reduced development costs.

## 1. Introduction

Integrated circuit (IC) packaging technology is continually innovating, with chips becoming lighter, thinner, and shorter. Due to the reduced size and an unmatched coefficient of thermal expansion (CTE) between the materials of the package, warpage occurring after completion of the thermal process can cause numerous problems. Large deformation results in the weak attachment of bumps or copper rods to the circuit board, which could even damage the structure and signal. Therefore, many studies have addressed warpage-related issues, and finite element (FE) simulation methods have been employed to analyze warpage behavior.

In the related literature on warpage theory because IC packages are composed of composite materials, the warpage behavior generated by thermal processes has been concluded to be complicated. Timoshenko [1] proposed a theory to explain the warpage caused by temperature changes in two bonded materials. Chen [2] discussed the effect of two bonding materials on warpage at different temperatures, theoretically analyzed a bimaterial structure, and sketched a multilateral structure. Garrett [3], of the technical department of Akrometrix, used Timoshenko’s biomaterial warpage theory [1] to derive the variables of warpage after an IC molding process: material properties (Young’s modulus and CTE), mold compound, substrate thickness, and temperature of the molding process. Wu et al. [4,5] discussed how the material properties of epoxy composites affected the performance of electronic devices and discovered the desirable dielectric and thermal properties for their design; they also studied the microwave absorption for nanorod and spinel structures [6,7]. Although some studies have analyzed package warpage theoretically, they simplified the structure into only the mold compound and substrate. A package also contains complex structures such as a die, wire bonding, copper rods, and bump solder balls. Thus, this simplification cannot fully reflect the actual situation.

Some scholars have recently employed FE simulation to analyze package warpage. Dudek et al. [8] used FE simulation to analyze the effects of the mechanical and thermal properties of the material body on warpage. In addition, Hu et al. [9] employed FE simulation to investigate plastic ball grid array packaging technology and developed a bimaterial warping model, where the substrate and mold compound in the package were used to discuss the effect of the molding process on warpage. Moreover, Huang et al. [10] used FE simulation to explore the effect of geometric die and substrate thicknesses on warpage. For packaging technology, they employed a single-sample window ball grid array, and they obtained simulated values consistent with the trend described by Timoshenko’s bimaterial theory [1] for the effect of the reflow process on warpage. Chae and Ouyang [11] discussed the effect of molding temperature on strip warpage for flip-chip strip packaging technology. They discovered that a high molding temperature will cause large strip warpage. In addition, they proposed the use of mechanics of composite materials theory for calculating the CTE of a substrate and mold compound at 25, 150, and 260 °C. Huber et al. [12] employed an FE simulation to determine the effect of mold compound on warpage after a long period of thermal aging. Bin et al. [13] applied FE simulation to fine pitch ball grid array packaging technology; they also discussed the effect of the geometric thickness of the mold compound and die on warpage when a strip was subjected to the molding process. The results revealed a negative correlation between the strip warpage and geometric thickness of the mold compound and die. Zheng et al. [14] proposed a reference temperature calibration of the flip-chip warping simulation model, obtaining consistency and small errors between the simulated and experimental warpage values. Chen et al. [15] employed FE simulation to investigate embedded silicon fan-out wafer-level package technology. The technique does not require EMC materials, and its structure is relatively simple. Therefore, the proposed simulation model achieved relatively good agreement with experimental and theoretical value, and the error between the experimental and the simulation results was only approximately 9%. However, this method is not feasible for a more complex simulation model with an EMC structure. Tsai et al. [16] proposed a new Suhir-solution-based theory for predicting the thermal deformation of flip-chip packages with the capillary underfill process and discussed the effect of the temperature of the reflow process on warpage. The deviation between the model and experimental results was approximately 25%. They also proposed a strain gauge measurement associated with a beam model theory to determine the thermally induced warpages of packages. The thermal strain results were consistent with those of validated FEM. Therefore, the strain gauge method proved feasible in determining the thermal warpages of packages [17]. Yao et al. [18] proposed an analytical model to evaluate the pore and superficial permeability of an underfill porous medium in a flip-chip packaging; they also presented an approach to predict the flow front and the filling time [19]. Chiang et al. [20] proposed an overview of artificial intelligence assisted design on simulation technology for reliability life prediction of advanced packaging. Developers only need to input geometric data of the package structures, and then the reliability life cycle can be obtained by this AI-trained model. Lin et al. [21] presented a finite element method to predict the final warpage of an ultra-thin flip chip scale package based on chemical shrinkage and cure-dependent viscoelasticity of molded underfill. Errors between the experimental and simulation results were approximately 10%. Cheng et al. [22] investigated the warpage behavior of a flip chip package-on-package (FCPoP) assembly during the fabrication process. They took some effects into account, such as the viscoelastic behavior and cure shrinkage of the epoxy molding compound. The results showed that simulation data fell within the ranges of the measured data.

The flip-chip is a commonly used packaging technique. Figure 1 shows the flip-chip process flow investigated in this study, and Figure 2 shows a schematic of a single flip-chip unit. As shown in Figure 1, the bonding process involves 12 steps:Wafer Grinding: the wafer is first processed by grinding before any fabrication procedure is conducted.Wafer Saw: the wafer is diced into small dies of target size.Flip-chip Bond: the cut die obtained from Step 2 is placed on the substrate with the help of bumps and soldering flux, but these are not completely fused together.Reflow: through the reflow process, the bumps and soldering flux on the substrate are fused so that the die can be fixed on the substrate.Flux Cleaning: plasma cleaning can remove contaminants formed during the production process, thereby effectively enhancing the strength of the bond between the die and substrate.Pre-MD Baking: this refers to the baking before molding, where water moisture subsequently formed in the die, substrate, and bumps must be completely removed to ensure that the mold compound fits tightly to protect the die, substrate, and bumps.Pre-MD Plasma Cleaning: the plasma surface is cleaned to remove impurities on the surface so that gaps between the internal components can be filled during molding.Molding: a mold compound is injected into the package to seal all the components, protecting the die and bumps inside the device.Post-molding cure (PMC): the sealed device is cured again to enhance its structural stability.Pre-treatment: pre-heat treatment before implantation of solder balls.Ball Mounting and Reflow: solder balls are implanted underneath the base substrate for future signal connection with the external circuit.Package Saw: the strip is diced into single wafers for packaging and shipping.

Because of the CTE mismatch between the packaging material, strip warpage often occurs during processes that require heating. Among the procedures, Steps 3–4, 6, and 8–11 are all performed at high temperatures. However, based on manufacturing experience, the strip is almost completely flat during Steps 3–7. No strip warpage occurs, even during the molding process in Step 8, when the strip is sealed by the mold compound and other materials at 175 °C. At this point, the strip remains almost flat. As discussed in several papers [3,9,11,14], strip warpage mainly occurs when the device is cooled to room temperature at 25 °C after the molding process. Severe strip warpage occurring during the post-molding period reduces product yield. The literature [8,9,10,11,12,13,14,20,21,22] indicates that FE simulation is frequently used to solve packaging warpage problems. Material parameters are crucial for determining whether consistency between simulated and experimental values is achieved. In particular, the mold compound is a high molecular polymer with a Young’s modulus and CTE that exhibit large temperature-based variations. In addition, few scholars have simultaneously simulated the effects of different geometric structures and process temperatures on warpage.

The present study input the temperature variation curve of the material properties into the simulation model and simulated the effects of different steps in the continuous process on warpage. The effects of the process temperature and geometric thickness of the mold compound, substrate, and die on warpage were also determined. Therefore, this study simulated a more realistic situation than previous studies. The proposed model and warpage analysis method can be used by designers to predict warpage under a continuous process and identify the optimal parameter design conditions for reducing strip warpage.

## 2. Research Method

The research method was divided into three steps: strip model, Taguchi method, and structural design.

### 2.1. Strip Simulation Model

#### 2.1.1. Model Establishment

This study used the COMSOL Multiphysics software to establish a strip model. Table 1 lists the specifications of the strip structure. The shape of the strip included a long side (x-direction) and a short side (y-direction). When the molding process was cooled to room temperature (25 °C), the amount of warpage on the short side of the strip was relatively small; only the direction of the long axis exhibited severe warpage (Figure 3). Table 2 lists the experimental values of the strip warpage (with compound 1). As shown in the table, no warpage occurred on either the long or short side during the molding process (Step 8) at 175 °C. However, as the strip was cooled to 25 °C, the CTE mismatch between the packaging materials caused a severe warpage of 7 mm but no measurable warpage on the short side. Until the end of Step 11 (ball mount), only minor warpage was observed on the short side. Therefore, strip warpage mainly occurred on the long side. To identify the main cause of warpage and reduce the simulation time required, this study simplified the 3D strip model to a 2D strip model. Subsequently, because a strip is a symmetrical structure, a quarter of the 2D strip model was used in this study. Figure 4 illustrates the quarter 3D strip model and 2D strip models. Figure 2 shows a structural diagram of a single unit in the strip, where each strip contained a total of 119 single-unit chips, and the structures included a mold compound, die, bump, and substrate. As shown in Table 3, we also validated the feasibility of a 2D simulation model. The results obtained from the 2D simulation model were found to be consistent with the experimental values, suggesting that it is possible to simplify the 3D model to a 2D model.

#### 2.1.2. Establishing Material Parameters

Material parameters are crucial to ensure that the simulation results of a model match the actual situation. As illustrated in Figure 2, four materials were present in the structure: bump, die, substrate, and mold compound. To match the materials on the production line, the bump was simulated as SAC405 (95.5Sn:4.0Ag:0.5Cu) and the die as being silicon. As shown in Table 4 and Table 5, the most important issue was based on [8,9,10,12]. The mold compound is a polymer material, whereas the substrate is composed of different materials, so it has varying mechanical and thermal properties due to differences in ambient temperature. The mold compounds employed in this study were the mold compounds 1 and 2, which are used on the production line. A dynamic mechanical analyzer (DMA) was employed to measure the Young’s modulus of the mold compounds and substrate. The range of temperatures used was 25–260 °C, and the Young’s modulus curves (E(T) curves) of the substrate and mold compounds are presented in Figure 5a and Figure 6a, respectively. The Young’s modulus curve is steep and has a negative slope at ambient temperatures of both mold compounds 1 and 2, close to its glass transition temperature (Tg; ca. 165 °C and 130 °C for mold compounds 1 and 2, respectively). A thermal mechanical analyzer (TMA) was employed to measure the CTE of the substrate and mold compounds 1 and 2 over the temperature range of 25–260 °C (Figure 5b and Figure 6b, respectively). The CTE curves of mold compounds 1 and 2 are steep and have a positive slope at ambient temperatures close to Tg. The CTE measurement results obtained for the substrate agree with the IPC-4101 specification [23]. Unlike related studies [8,9,10,11,12,13,14], this study input the temperature variation curves E(T) and α(T) of mold compounds and the substrate into the simulation model, which ensured that the material parameters of the simulation model were close to the actual situation.

As illustrated in Figure 2, the bump was located between the die and the substrate and was used for signal connection. According to the mechanics of materials [24], when the stress of a material exceeds the yield stress, the material is no longer a linear elastic material, and it undergoes plastic deformation. The stress–strain diagram of an SAC405 bump was presented in [25]. The stress–strain diagram obtained in the present study (Figure 7) indicates a yield stress of 26 MPa and was input into the simulation model. According to [26], a creep effect can occur in a metal when the ambient temperature exceeds one-third of its melting point, and one-third of the melting point of an SAC405 bump is 72.28 °C. The flip-chip considered in this study had a process temperature of ≥175 °C. This study was different from other related studies [8,9,10,11,12,13,14] as the plastic effect of a bump [25] and the creep effect [26] were considered in this model. Both effects were input into the simulation model to match a realistic situation.

#### 2.1.3. Boundary Condition Settings

The process flow of the flip-chip on the production line consisted of twelve steps, as illustrated in Figure 1. The issue was warpage generation due to the unmatched CTEs of the package materials. Warpage mostly occurred after thermal processes. The six steps involving thermal processes were reflow, pre-MD (molding) baking, molding, PMC, pretreatment, and ball mounting. However, according to experience from production line workers, strips are almost flat after Steps 3–7, as illustrated in Figure 8a. Even when the mold compound is combined with the other materials into the strip package at 175 °C during molding, warpage had not yet occurred, and the strip was almost flat. However, severe warpage occurred when the strip was cooled to a room temperature of 25 °C, as shown in Figure 8b. In related studies [8,9,10,11,12,13,14], the simulation model was set for a stress-free state at a molding temperature of 175 °C. As illustrated in Figure 8, the stage and strip at both ends of the strip model were defined as contact points at position A, and the stage was a simple support. For any end point A located between y = 0 and w (where w is the width of the strip), the following 2D boundary condition can be imposed: point A is allowed to move freely along the x-direction with an arbitrary displacement (displacement along the x-direction is u_x_ = constant) but fixed along the z-direction (displacement along the z-direction is u_z_ = 0).

According to [8,9,11,13,14], warpage mainly occurs during molding. Therefore, the simulation model starts from the molding process (Step 8). Continuous simulation calculations of different processes were conducted to simulate Steps 8–11 of the thermal process. Table 6 details the process temperature ranges and times for these steps. First, the molding process was simulated (Step 8) by cooling the system from 175 °C to a room temperature of 25 °C. Subsequently, the temperature was increased to and then maintained at 175 °C for 240 min during the PMC process (Step 9) to completely cure the mold compound and eliminate internal stress. Finally, in Steps 10 and 11, the reflow process for pretreatment of the solder ball and implantation of the ball into the substrate was simulated.

### 2.2. Taguchi Method

Various variables, such as geometric structure and process temperature, can affect the warpage during the process. Few scholars have simultaneously studied the effects of different geometric structures and process temperatures on strip warpage. Therefore, this study explored the effects of process temperature and geometric thickness of the mold compound, substrate, and die on strip warpage. The process temperature was changed mostly in Steps 8 and 9. Hence, this study used Taguchi’s orthogonal arrays to establish an L16 variable combination. Table 7 lists the control factors and their settings. This study explored five factors: three related to structural thickness (thickness of the mold compound, die, and substrate) and two related to process temperature (molding and PMC temperatures).

### 2.3. Structural Design

IC packages are composed of numerous different materials and geometrical shapes. This study used the neutral axis theory of composite materials [24] as the structural design criteria. Severe strip warpage occurs when the strip is cooled to room temperature (25 °C). Therefore, this study investigated the relationship between the *z_n_*-coordinate of the neutral axis and warpage at 25 °C. Equation (1) was used to calculate the neutral axis z-coordinate formula of the composite material, where *A_Mold compound_*, *A_Die_*, *A_Bump_*, and *A_Substrate_* are the areas of the mold compound, die, bump, and substrate, respectively. Table 8 lists the Young’s modulus of each material at 25 °C. The Young’s moduli were normalized by dividing each by the minimum. *n* represents the transformation factor, and *n_Mold compound_*, *n_Die_*, *n_Bump_*, and *n_Substrate_* represent the proportional Young’s moduli of the materials. Furthermore, *z_Mold compound_*, *z_Die_*, *z_Bump_*, and *z_Substrate_* are the centroid z-coordinates of the mold compound, die, bump, and substrate, respectively. Substituting the values in Table 8 into Equation (1), the z-coordinate *z_n_* of the neutral axis of the 2D strip in Figure 3 was obtained as 280.61 µm.
(1)$$z_{n}=\frac{A_{1}\times n_{1}\times z_{1}+A_{2}\times n_{2}\times z_{2}+A_{3}\times n_{3}\times z_{3}+A_{4}\times n_{4}\times z_{4}}{A_{1}\times n_{1}+A_{2}\times n_{2}+A_{3}\times n_{3}+A_{4}\times n_{4}}\;\;\;\;\;$$
were
*A*_1_ = *A_Mold compound_, n*_1_ = *n_Mold compound_, z*_1_ = *z_Mold compound_*.*A*_2_ = *A_Die_, n*_2_ = *n_Die_, z_2_* = *z_Die_*;*A*_3_ = *A_Bump_, n*_3_ = *n_Bump_, z*_3_ = *z_Bump_*;*A*_4_ = *A_Substrate_, n*_4_ = *n_Substrate_, z*_4_ = *z_Substrate_*.

## 3. Results

### 3.1. Experimental Results

Figure 9 indicates eight points (D1–D8) as the measuring positions along the strip. D4 and D8 were in the middle of the long side of the strip, while D2 and D6 were in the middle of the short side of the strip. A ruler was used to measure the warpage at the start of the molding process at 175 °C and the end of the process at 25 °C. Table 9 lists the experimental values of strip warpage at the reference points during the molding process. The results show that D4 and D8 are the positions at which the maximum warpage occurred. A similar process was applied to obtain the maximum warpage of the strip during the post-mold curing, pre-treatment, and ball mount processes. These results are listed in Table 10.

### 3.2. Simulation Results

#### 3.2.1. Experimental and Simulation Results

Most related studies only considered a single process step (molding and reflow) [8,9,10,11,12,13,14]; in contrast, this study successfully simulated the continuous flip-chip process from molding to ball mounting (Steps 8–11). Table 10 presents a comparison of the simulation results and experimental values, and Figure 10 illustrates the simulation results of strip warpage after each process for mold compound 1. The warpage trends are consistent, revealing a concave shape facing downwards. The simulated and experimental values were similar, with differences all lower than 8%. Therefore, the model established in this study is feasible for simulating the flip-chip process.

#### 3.2.2. Comparison of Mold Compounds

To investigate the effect of mold compounds for strip warpage on the production line, the strip warpage of different mold compounds (mold compounds 1 and 2) was compared under the same geometry and process conditions. Table 11 presents a comparison of strip warpage for mold compounds 1 and 2, and the results demonstrate that the strip containing mold compound 2 warped less than mold compound 1, regardless of the specific process step. The two main factors affecting warpage under the same geometric structure and process conditions are as follows:The CTE difference between the mold compound and substrate had a greater impact, the larger the difference, the larger the strip warpage.The Young’s modulus of the mold compound exerted an effect, the larger the young’s modulus, the greater the structural rigidity of the strip and the lower the warpage.

As shown in Figure 5b and Figure 6b, the CTE difference between mold compound 2 and the substrate was smaller than that between mold compound 1 and the substrate. In addition, Figure 6a indicates that the Young’s modulus of mold compound 2 was slightly larger than that of mold compound 1. The use of mold compound 2 resulted in less strip warpage for each process than the use of mold compound 1.

## 4. Discussion

### 4.1. Taguchi Analysis

Since the degree of warpage that occurs during ball mounting (Step 11) directly affects the single-chip yield in the singulation step (Step 12) of the flip-chip process, this study focused on the strip warpage that occurred during the ball mounting process. Figure 11 displays the Taguchi analysis’ main effect diagram of the ball mounting warpage. According to Figure 11, the main factors affecting strip warpage are the die and substrate thickness, followed by mold compound thickness and molding temperature. The least influential factor is PMC temperature. This study investigated the effects of these five factors sequentially. First, the effect of die thickness was evaluated. The Young’s modulus of the die was relatively high (131 GPa), which indicates that the die was the most rigid material in the strip. Therefore, when the die thickness was increased, the structural rigidity of the strip notably increased, reducing the amount of warpage. Second, the effect of substrate thickness factor was determined. When the substrate thickness was increased to 300–500 μm, the amount of warpage rose sharply to >7 mm; thus, substrate thickness was positively correlated with warpage (Figure 11). Third, the effect of mold compound thickness was investigated. A higher mold compound thickness resulted in greater strip rigidity. However, the warpage was considerably reduced only when the thickness was increased to 1100 μm; thicknesses of 150, 450, and 750 μm were unable to cause large warpage reduction. Fourth, the effect of molding temperature factor was evaluated. The smaller difference between the molding process temperature and room temperature of 25 °C resulted in the lower warpage. Nonetheless, the molding process temperature must be higher than the Tg of the mold compound. Finally, the effect of PMC temperature factor was determined. The results clearly demonstrate that a change in the PMC process temperature has no strong effect on warpage.

For the current flip-chip structure design, the following four process condition designs can reduce the amount of strip warpage:greater die thickness (>150 μm).greater mold compound thickness (>1100 μm).smaller substrate thickness (<100 μm); andlower molding temperature, although it should not be lower than the Tg of the mold compound.

### 4.2. Structural Design Criteria

According to the Taguchi analysis presented in Section 4.1, a greater die thickness, greater mold compound thickness, smaller substrate thickness, and lower molding temperature are ideal for reducing warpage. However, the trend in manufacturing is for thinner, lighter, and shorter chips, and increasing the die and mold compound thicknesses to reduce warpage is not optimal. Therefore, this study constructed a structural strip design based on composite material neutral axis theory [24,27] and investigated the relationship between z_n_ and warpage. Table 12 lists the ball mounting warpage simulation results, where the original parameters reflect the original conditions on the production line, and z_n_ is the neutral axis z-coordinate of each structural condition. The positions of the structural neutral axis z_n_-coordinates were divided into three categories: (1) neutral axis on the mold compound (No. 13), (2) neutral axis on the die (Nos. 2, 3, 4, 5–12, and 14–16), and (3) neutral axis on the bump (No. 1).

First, the neutral axes on the mold compound (No. 13) and die (Nos. 2, 3, 4, 5–12, and 14–16) are discussed. Three characteristics were identified after comparing with the main effect analysis diagram in Figure 11:less warpage occurred when the substrate thickness was <180 μm (e.g., Nos. 16, 12, 6, 11, and 15) and the warpage was <5.15 mm.greater warpage occurred when the substrate thickness was ≥300 μm (e.g., Nos. 14, 7, 9, and 10) and the warpage was >7.92 mm; andNos. 5 and 8 were special because the substrate thickness was only 180 μm (in No. 5), and the die thickness was only 75 μm, which caused insufficient structural rigidity, resulting in a warpage of 7.63 mm.

In contrast, although the substrate and die thicknesses of No. 8 were 300 and 400 μm, respectively, the structural rigidity was sufficient, and thus the warpage was only 4.75 mm. For No. 1 (bump on the neutral axis), even if the mold compound and die thicknesses were only 150 and 75 μm, respectively, with the substrate thickness being only 100 μm, a relatively small warpage of 4.69 mm occurred. Therefore, the group with neutral axis coordinates below the die and above the bump resulted in relatively small warpage.

In summary, the die and substrate thicknesses have a strong effect on warpage in the structural design. This study discovered that in the flip-chip process design, in addition to meeting the production requirement that the mold compound should nearly completely cover the die, two conditions must be fulfilled to minimize the amount of strip warpage: (1) neutral axis on the bump and (2) neutral axis on the die with a die thickness of >150 μm and substrate thickness of <180 μm. If one of these conditions is met, the strip will have a smaller warpage of <5.15 mm. The authors suggest that designers set the neutral axis of the strip structure on the bump, which is more suitable for the current production trend of thin, light, and short design.

## 5. Conclusions

This study successfully established a strip warpage simulation model of the flip-chip process and investigated the effects of structure design and process (molding, PMC, pretreatment, and ball mounting) on strip warpage. The errors between model and experimental values were less than 8%, indicating that the simulation method can be applied to the flip-chip process steps and can be extended to strip warpage analysis of different mold compounds in the future. In addition, Taguchi analysis was employed to identify the key factors affecting strip warpage, which were discovered to be die thickness and substrate thickness, followed by mold compound thickness and molding temperature. Although greater die and mold compound thicknesses result in less warpage, they cause a substantially greater overall strip thickness, which does not comply with the current trend towards thin, light, and short chips. To overcome this problem, this study proposed the design concept of setting the neutral axis of the strip structure on the bump, which reduces the amount of strip warpage and the overall strip thickness. In summary, the model proposed in this study can be used to evaluate the effect of structural design and process parameters on strip warpage and can provide strip design guidelines for minimizing strip warpage to requirements of the production line. Moreover, the guidelines can accelerate the product development cycle and improve product quality with reduced development costs.

## Figures and Tables

**Figure 1 materials-15-00323-f001:**
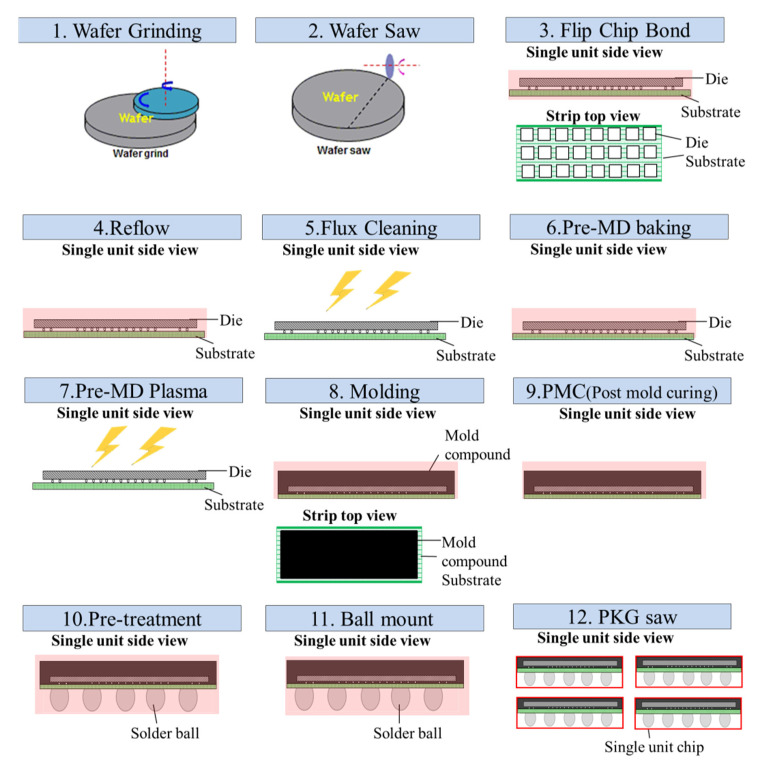
Flip-chip process flow.

**Figure 2 materials-15-00323-f002:**
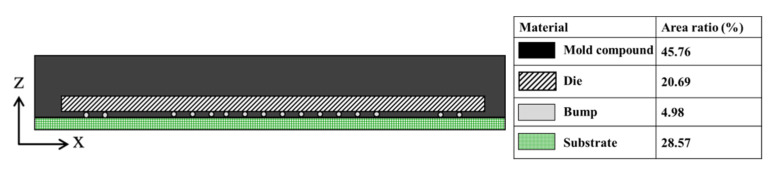
Structural schematic of a single unit inside a strip.

**Figure 3 materials-15-00323-f003:**
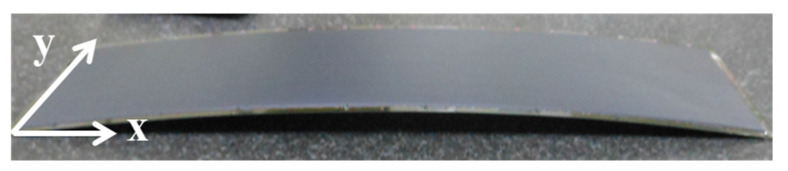
Actual strip warpage on the production line.

**Figure 4 materials-15-00323-f004:**
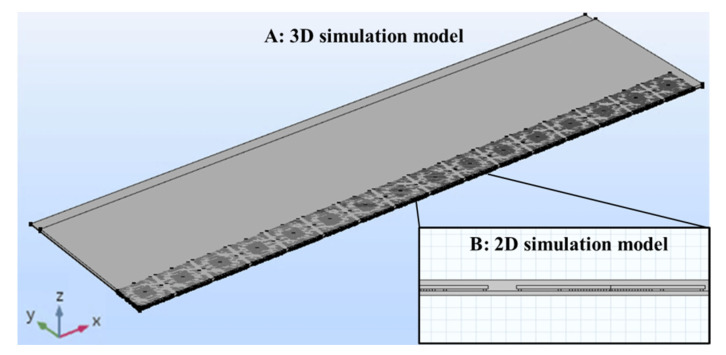
One quarter of the (**A**) 3D model and (**B**) 2D model.

**Figure 5 materials-15-00323-f005:**
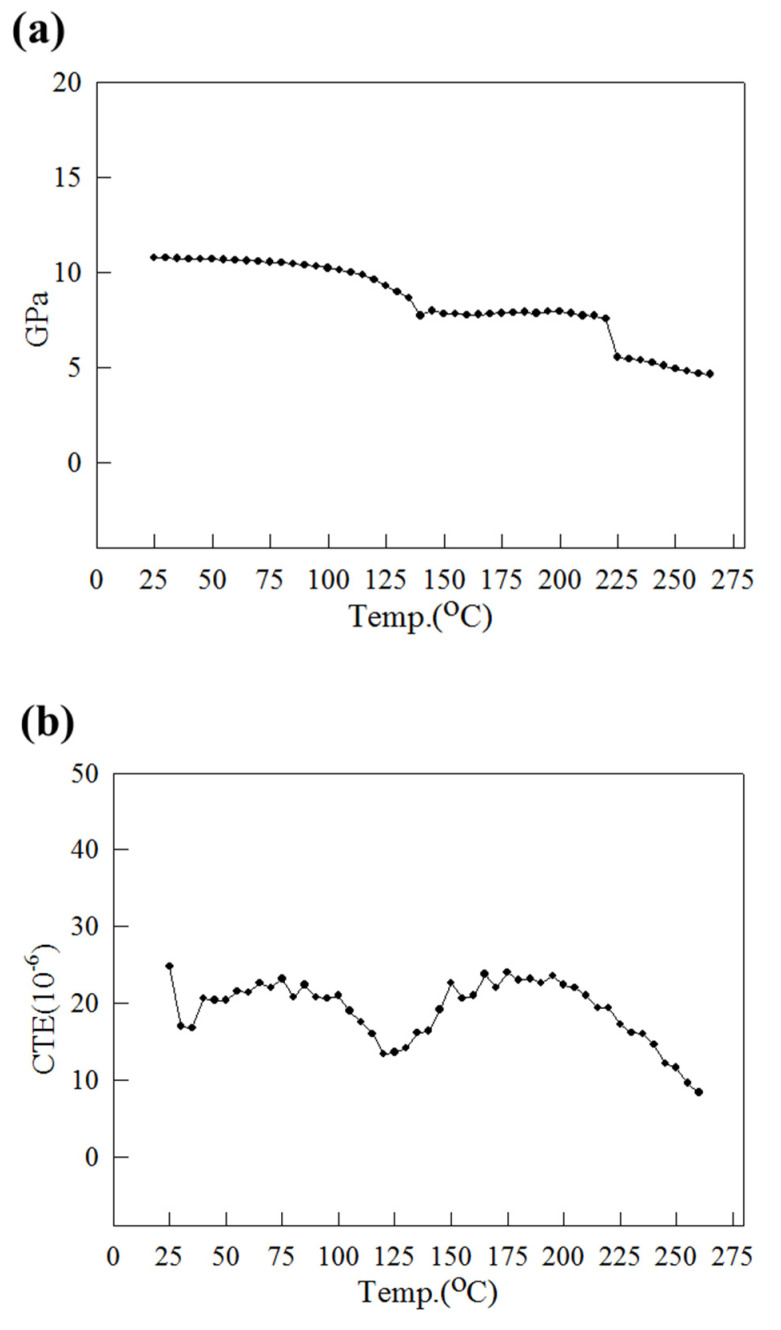
(**a**) Young’s modulus E(T) and (**b**) CTE α(T) of substrate.

**Figure 6 materials-15-00323-f006:**
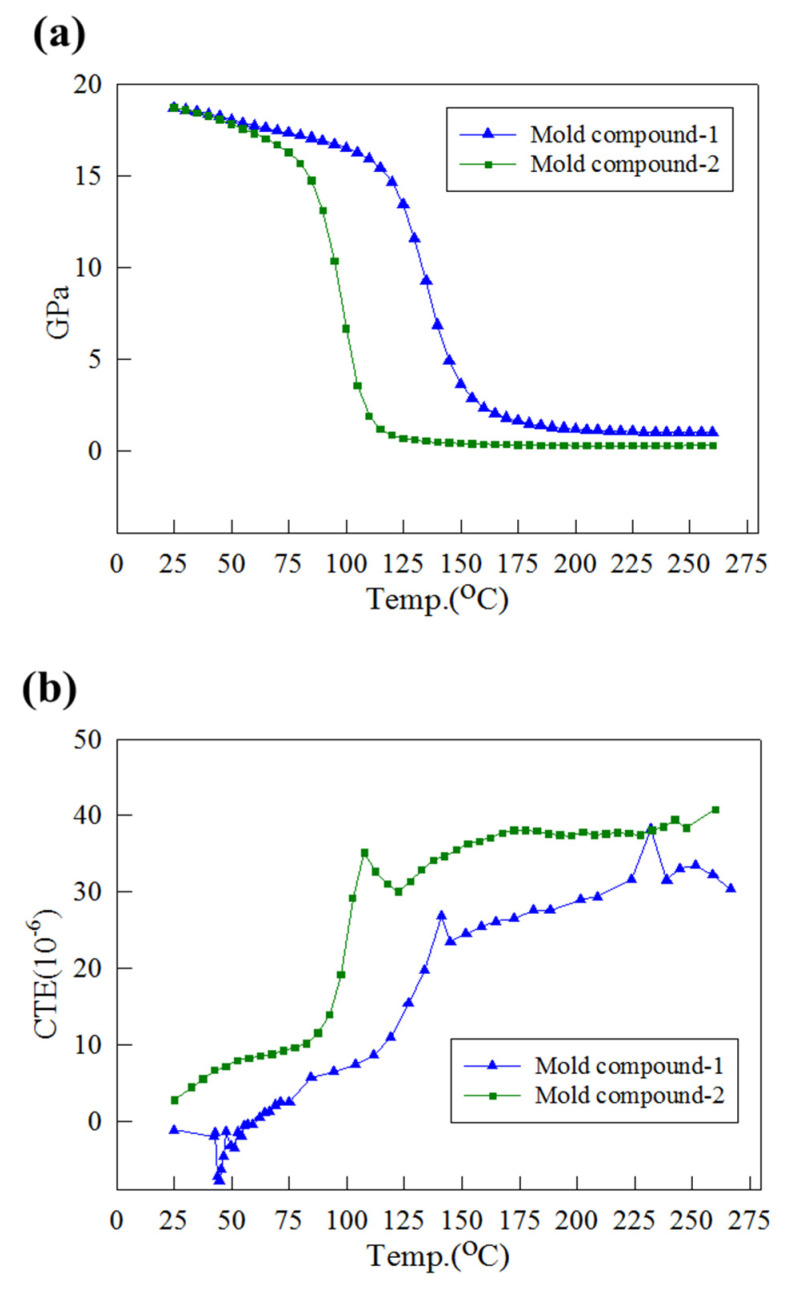
(**a**) Young’s modulus E(T) and (**b**) CTE α(T) of mold compounds.

**Figure 7 materials-15-00323-f007:**
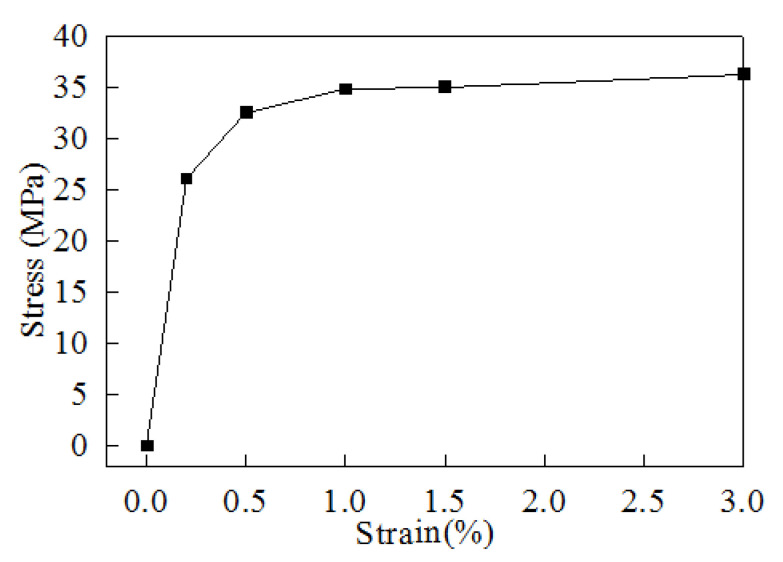
Stress–strain curve of the SAC405 bump [25].

**Figure 8 materials-15-00323-f008:**
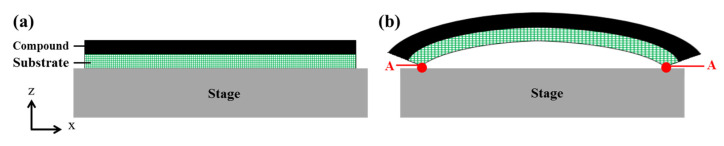
Strip placed on the stage: (**a**) at 175 °C after molding and (**b**) after cooling to 25 °C.

**Figure 9 materials-15-00323-f009:**
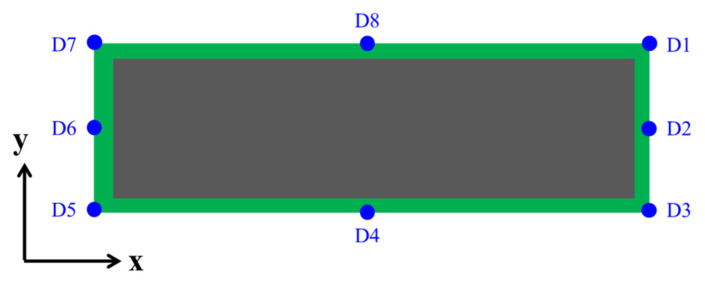
Measuring positions along the strip (D1–D8).

**Figure 10 materials-15-00323-f010:**
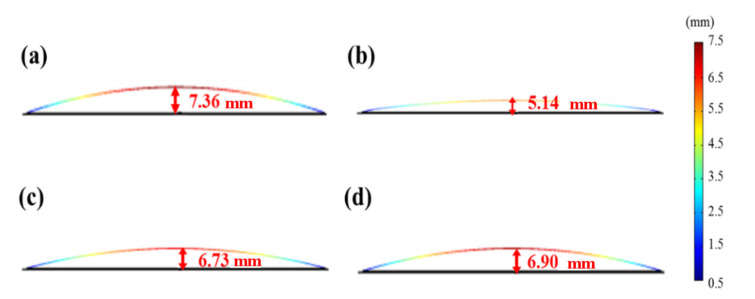
Strip warpage simulation results for mold compound 1: (**a**) Step 8—molding, (**b**) Step 9—PMC, (**c**) Step 10—pretreatment, and (**d**) Step 11—ball mounting.

**Figure 11 materials-15-00323-f011:**
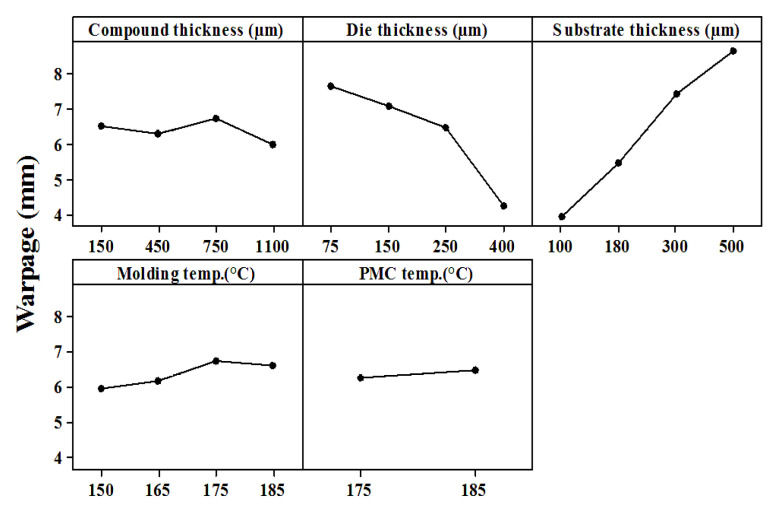
Main effect diagram of factors affecting warpage during ball mounting.

**Table 1 materials-15-00323-t001:** Specifications of the strip structure.

PKG Information	
PKG size (mm^2^)	7 × 7
Mold compound thickness (µm)	450
Die size (mm^2^)	6.3 × 6.34
Die thickness (µm)	150
Bump type	SAC405
Bump pitch (µm)	190

**Table 2 materials-15-00323-t002:** Experimental values of strip warpage.

Process Flow		Strip Warpage (mm)	
		Long Side (x-Direction)	Short Side (y-Direction)
8. Molding	175 °C	0	0
	25 °C	7	0
9. PMC		5	N/A
10. Pre-treatment		7	N/A
11. Ball mount		7.5	N/A

**Table 3 materials-15-00323-t003:** Experimental values vs. simulation values of strip warpage.

Process Flow		Strip Warpage (mm)	
		Experimental Value	Simulation Value
8. Molding	175 °C	0	0
	25 °C	7	7.36
9. PMC		5	5.14
10. Pre-treatment		7	6.73
11. Ball mount		7.5	6.90

**Table 4 materials-15-00323-t004:** Material parameters of bump, silicon, and substrate.

	Bump-SAC405	Die (Silicon100)	Substrate
Young’s modulus (GPa)	53	131	Figure 5a
Poisson’s ratio	0.40805	0.27	0.2
Density (kg/m^3^)	7445.45	2330	1938
CTE (ppm/°C)	20	2.8	Figure 5b

**Table 5 materials-15-00323-t005:** Material parameters of mold compounds.

	Mold Compound 1	Mold Compound 2
Young’s modulus (GPa)	Figure 6a	Figure 6a
Poisson’s ratio	0.3	0.3
Density (kg/m^3^)	2010	1990
CTE (ppm/°C)	Figure 6b	Figure 6b

**Table 6 materials-15-00323-t006:** Process temperature ranges and times for flip-chip.

Process Flow	Temperature Range (°C)	Process Time (s)
8. Molding	175 → 25	30
9. PMC	25 → 175 → 25	20040
10. Pre-treatment	25 → 238 → 25	935
11. Ball mount	25 → 238 → 25	935

**Table 7 materials-15-00323-t007:** Control factors and their settings.

Name of Control Factors	Level 1	Level 2	Level 3	Level 4
Mold compound thickness(µm)	150	450	750	1100
Die thickness (µm)	75	150	250	400
Substrate thickness(µm)	100	180	300	500
Molding temp. (°C)	150	165	175	185
PMC temp. (°C)	175	185		

**Table 8 materials-15-00323-t008:** Transformation factor *n* for each material.

Material	Young’s Modulus (GPa)	*n* = Material (E)/Substrate (E)
Substrate	10.75	1
Bump	53	4.93
Die	131	12.18
Mold compound 1	18.66	1.73

**Table 9 materials-15-00323-t009:** Experimental values of strip warpage at reference points.

Process Flow		Strip Warpage (mm)							
		Long Side (x-Direction)		Short Side (y-Direction)		Corner			
		D4	D8	D2	D6	D1	D3	D5	D7
8. Molding	175 °C	0	0	0	0	0	0	0	0
	25 °C	7	7	0	0	0	0	0	0

**Table 10 materials-15-00323-t010:** Comparison of strip warpage simulation and experimental values using mold compound 1.

Process Flow	Strip Warpage (mm)		
	Experimental Value	Simulation Value	Error (%)
8. Molding	7	7.36	5.14
9. PMC	5	5.14	2.8
10. Pre-treatment	7	6.73	3.85
11. Ball mount	7.5	6.90	8.00

**Table 11 materials-15-00323-t011:** Comparison of strip warpage simulation values for mold compounds 1 and 2.

Process Flow	Strip Warpage (mm)	
	Mold Compound 1	Mold Compound 2
8. Molding	7.36	5.02
9. PMC	5.14	3.48
10. Pre-treatment	6.73	4.65
11. Ball mount	6.90	4.71

**Table 12 materials-15-00323-t012:** Warpage simulation results obtained with production-line and L16 parameters.

No.	Mold Compound Thickness (µm)	Die Thickness (µm)	Substrate Thickness (µm)	Molding Temp. (°C)	PMC Temp. (°C)	Ball Mount Warpage (mm)	z_n_ (µm)
Original	450	150	180	175	175	6.36	280.61
1	150	75	100	150	175	4.69	156.18
2	150	150	180	165	175	6.21	258.21
3	150	250	300	175	185	7.71	410.69
4	150	400	500	185	185	7.47	661.20
5	450	75	180	175	185	7.63	264.51
6	450	150	100	185	185	4.47	205.25
7	450	250	500	150	175	8.31	602.85
8	450	400	300	165	175	4.75	479.12
9	750	75	300	185	175	9.36	434.78
10	750	150	500	175	175	9.79	620.90
11	750	250	100	165	185	4.81	276.04
12	750	400	180	150	185	2.94	394.61
13	1100	75	500	165	185	8.98	703.36
14	1100	150	300	150	185	7.92	524.47
15	1100	250	180	185	175	5.15	421.30
16	1100	400	100	175	175	1.91	373.81

## Data Availability

Not applicable.

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
