# Peer review of "Study on the Strip Warpage Issues Encountered in the Flip-Chip Process"

_materials, 2022, doi:10.3390/ma15010323_

Round 1
Reviewer 1 Report
Comment 1:
Experiment details need to be given. Factors and their levels. Experimental procedure.
Comment 2:
The error between the experiment and simulation is 8%, too large. Needs to do uncertainty analysis, see the uncertainty analysis in the flip chip system - “A Surface Energy Approach to Developing an Analytical Model for the Underfill Flow Process in Flip-Chip Packaging” – j of electronic package.
Comment 3:
The current process looks like reflow process. The popular method is a dispensing process, I.e., the underfill material is filled into the clearance between the chip and board, which includes an arrange of solder bumps, see the literature “A Further Study on the Analytical Model for the Permeability in Flip-Chip Packaging”. The authors may need to comment on that situation to clarify the scope of the work.
Comment 4:
Originality of the work needs to be clearly stated. It is known that there are a lot of finite element modelling for such a system.
Comment 5:
Whether the non-uniform underfill may affect the warping?
Comment 6:
The paper appears not too much fit to the scope of the journal.
Reviewer 2 Report
The work is written well. It has a research aspect and its results may be applied in the future. Careful development of the process may improve the quality of the product and reduce the final price. I have no comments to the reviewed work. I propose to publish the work without corrections.
Reviewer 3 Report
The article is about strip warpage issues encountered in the flip-chip process. However, the article is lack of novelty due to the fact that the authors disregards the scientific literature from last 5 years (!!!!), except only 2 citations. Without a consistant state-of-the-art of the subject, entire research cannot be considered being original and new. The article is not suitable for publication.
Reviewer 4 Report
This study successfully established a strip warpage simulation model of the flip-chip process and investigated the effects of structural design and process (molding, post-mold curing, pretreatment, and ball mounting) on strip warpage. However, I considered it can be published in our journal after some revision:
- Obvious grammar and spelling mistakes could be easily found in the manuscript, and some of the mistakes are listed below:
Page1 Keywords: “flip-chip process; strip warpage; bump”. The author should examine the manuscript carefully and revise it carefully. For example, the first letter should not be capitalized and the font size should be uniform.
I suggest that the authors improve the English presentation and polish it with professional native speakers of English.
- There is no need to put a comma after formula 1 on page 9. Figure 9 is not aligned. The abscissa of the second row of Figure 10 should be set at the same interval. "9." in Table 10 should be adjusted to make the table spacing as uniform as possible. The "Zn" font in Table 11 is too small and should be consistent with other font sizes.
- Some recent work of related materials can be compared with the present work such as “Nano-Micro Letters, 2021, 13: 175., Nanomaterials, 2021, 11(8): 1898., Chemical Engineering Journal 2021, 420:129907., Journal of Materials Science: Materials in Electronics, 2021, 32(16), 20973-20984. https://doi.org/10.1016/j.jmst.2021.06.034., DOI: 10.1007/s42114-021-00304-2.”
- The line in picture 5 and picture 6 is too thin and the type is too small. It should be adjusted to make it clearer. There are some mistakes in Table 4 and Table 5. Please check them again and make corrections.
- I suggest that the author amend post mold curing for post-mold curing in the manuscript. The author should pay attention to the uniformity of the manuscript format. All illustrations in a manuscript should be centered.
- The author should check the manuscript carefully for mistakes, pay attention to the logic of the article, and indicate the innovation of the article.
Round 2
Reviewer 1 Report
I am fine with the revision.
Reviewer 3 Report
After all improvements identified on the manuscript, the article is suitable for publication.